# Social drivers and pediatric injury outcomes in Northern Tanzania: A prospective pediatric injury registry secondary analysis

Natalie J. Tedford[1]*, Modesta Mitao[2], Timothy Antipas Peter[3,4], Linda Minja[3,4], Gabriele Nascimento de Oliveira[5,6], Raven Mingo[7], Getrude Nkini[3,4], Catherine Staton[6,8,9], Joao R. N. Vissoci[6,8,9], Blandina T. Mmbaga[2,3,4☯], Elizabeth M. Keating[6,10☯]*

1 University of Wisconsin School of Medicine and Public Health, BerbeeWalsh Department of Emergency Medicine, Madison, Wisconsin, United States of America, 2 Kilimanjaro Clinical Research Institute, Moshi, Tanzania, 3 Kilimanjaro Christian Medical Centre, Moshi, Tanzania, 4 Kilimanjaro Christian Medical University College, Moshi, Tanzania, 5 State University of Maringá, Graduate Program in Health Sciences, Health Sciences Center, Maringá, Paraná, Brazil, 6 Duke University Medical Center, Global Emergency Medicine Innovation and Implementation (GEMINI) Research Center, Durham, North Carolina, United States of America, 7 Mississippi Band of Choctaw Indians, Choctaw, Mississippi, United States of America, 8 Duke University Medical Center, Department of Emergency Medicine, Durham, North Carolina, United States of America, 9 Duke University, Duke Global Health Institute, Durham, North Carolina, United States of America, 10 University of Utah, Department of Pediatrics, Division of Pediatric Emergency Medicine, Salt Lake City, Utah, United States of America

☯ These authors are contributed equally and co-senior authors to this work.
* natalie.j.tedford@gmail.com (NJT); elizabeth.keating@hsc.utah.edu (EMK)

## Abstract

Trauma remains a leading cause of preventable childhood mortality and disability worldwide, with over 95% of pediatric injury-related deaths occurring in low- and middle-income countries (LMICs). Social drivers of health (SDH) – including socio-economic status, insurance coverage, and community infrastructure – strongly influence these outcomes but remain underexplored in sub-Saharan Africa. To examine the association between selected SDH factors and morbidity and mortality among injured children presenting to a tertiary referral hospital in Northern Tanzania. This secondary analysis utilized data from a prospective pediatric injury registry at Kilimanjaro Christian Medical Center (KCMC) from November 2020 to January 2024. Children under 18 years presenting with acute injuries were enrolled. SDH variables included residence type, payment method, insurance status, household composition, food security, and transfer status. Outcomes were in-hospital mortality and morbidity, defined using the Glasgow Outcome Scale – Extended Peds (GOS-E Peds). Multivariable logistic regression and modified Poisson regression models were used to identify independent predictors of mortality and morbidity. A total of 877 children were included (median age 7 years, 63% males). Mortality rate was 7.0%, while 38.8% experienced poor morbidity outcomes (GOS-E Peds ≥3). In multivariable analysis, lack of health insurance (adjusted OR=2.4, 95%CI 1.1—5.3) and transfer

**Data availability statement:** Public deposition of the data would breach compliance with the protocol and data-sharing agreement approved by our research ethics boards; therefore, we are ethically unable to share the data publicly. A data-sharing agreement with KCMC covers the data for this manuscript; thus, we are unable to share the data openly. For data access requests, please contact our non-author KCMC representative, Gwamaka William, at gwamakawilliam14@gmail.com.

**Funding:** The data for this paper came from work supported by a Fogarty Global Health Fellowship funded by the Fogarty International Center of the National Institutes of Health (grant number D43 TW009337). EMK receives funding from the Eunice Kennedy Shriver National Institute of Child Health and Human Development (grant number K23 HD112548). The funders had no role in study design, data collection and analysis, decision to publish, or preparation of the manuscript.

**Competing interests:** The authors have declared that no competing interests exist.

by ambulance to KCMC (adjusted OR=3.8, 95%CI 1.5—9.4) were associated with higher odds of mortality. Older age (adjusted PR = 1.05, 95%CI 1.02—1.09) predicted greater morbidity, while non-ambulance transfer was protective (adjusted PR = 0.74, 95%CI 0.56—0.97). Food insecurity remained an independent correlate of poor outcomes. Social and economic inequities, particularly inter-facility ambulance transfer, lower MUAC, and food security, were independently associated with adverse pediatric injury outcomes. Food insecurity and lack of insurance remained cross-cutting vulnerabilities. Integrating SDH surveillance into trauma care systems and addressing access barriers may reduce injury-related morbidity and mortality in LMICs.

## Introduction

Injuries are a leading cause of death in children aged 5–14 years globally, with low- and middle-income countries (LMICs) accounting for more than 95% of these fatalities [1]. Children in sub-Saharan Africa experience disproportionately higher injury risks due to structural, environmental, and social disparities, including limited access to healthcare, poverty, and inadequate safety infrastructure [2–4]. While biomedical factors influencing pediatric trauma outcomes are well documented, the contribution of social drivers of health (SDH) – the conditions in which individuals are born, live, and grow [5] – remains insufficiently characterized in LMICs.

SDH encompass domains such as economic stability, education, neighborhood environment, food security, and healthcare access [5]. These drivers shape exposure to risk, injury severity, and recovery trajectories [2,4]. In Tanzania and other LMICs, however, SDH variables are rarely integrated into injury registries or outcomes surveillance systems [6–11]. This limits the capacity to identify vulnerable groups and design context-specific interventions that improve post-injury outcomes. Prior studies have primarily focused on the incidence of pediatric injuries, whereas little is known about how post-injury trajectories, including survival and recovery, are influenced by socioeconomic inequities. Understanding these pathways aligns with the Sustainable Development Goals (SDG 3: Good Health and Well-Being; SDG 10: Reduced Inequalities), which emphasize equitable access to healthcare and injury prevention in children [9,12].

To address this gap, we conducted a secondary analysis of data from a prospective pediatric injury registry at a tertiary referral hospital in Northern Tanzania. Our objective was to examine the associations between selected SDH factors, including insurance status, community residence, transfer patterns, and food insecurity, and the mortality and morbidity outcomes of injured children. This work provides evidence to inform equitable injury care policies and future SDH monitoring in low-resource trauma systems.

## Materials and methods

### Ethics statement

This study was approved by the Kilimanjaro Christian Medical College (KCMC) Research Ethics and Review Committee (CRERC 2938), the National Institute for

Medical Research (NIMR/HQ/R.8a/Vol. IX/3889), and the University of Utah Institutional Review Board (IRB_00156728). Because this was a secondary analysis of de-identified registry data, all committees waived the requirement for individual consent. The pediatric injury registry from which data were drawn obtained verbal consent from the parent or guardian of each child at enrollment in accordance with national ethical standards.

## Study design

This was a secondary data analysis of a prospective pediatric injury registry conducted at KCMC, a tertiary referral hospital in Northern Tanzania. The registry was designed to inform pediatric trauma care quality improvement and systematically record demographic, clinical, and outcome data. For the present analysis, data collected between November 1, 2020, and January 31, 2024, were used.

## Study setting

KCMC serves as a zonal referral hospital for five northern Tanzanian regions — Arusha, Dodoma, Kilimanjaro, Singida, and Tanga — with a combined catchment population exceeding 12 million. The Emergency Medicine Department (EMD) receives approximately 1,400–1,700 pediatric patients annually, including those transferred from district hospitals and rural health centers. Pediatric trauma care is provided collaboratively by general surgeons, orthopedic surgeons, and pediatricians. KCMC also operates a specialized burn center, but Tanzania lacks a formal emergency medical services (EMS) system, and interfacility transfers are often conducted by non-specialized vehicles.

## Definitions

An "injury" was defined per the World Health Organization (WHO) as physical harm caused by acute exposure to mechanical, thermal, electrical, or chemical energy exceeding physiological tolerance thresholds [13]. Eligible injury mechanisms included fractures, burns, lacerations, road traffic injuries, traumatic brain injury, falls, drowning, poisoning, and animal bites.

Food insecurity was defined as caregiver-reported inability to provide ≥3 daily meals for the child, representing a household-level measure of food access and availability.

## Study participants

Children aged under 18 years presenting to the KCMC EMD with acute injury were enrolled into the pediatric injury registry at first presentation. Patients presenting for follow-up or chronic wound care were excluded. The present analysis included all registry patients with complete outcome data through hospital discharge.

## Study outcome variables

The primary outcomes were mortality (defined as in-hospital death following injury) and morbidity, measured at discharge using the Glasgow Outcome Scale – Extended Pediatrics (GOS-E Peds) [14]. GOS-E Peds categorizes outcomes from 1 (Good recovery) to 8 (Death) [14]. For analysis, scores of ≥3 represented "poor outcomes," indicating moderate disability, severe disability, or death, while ≤2 represented "good recovery.

For this analysis, variables were extracted from the Research Electronic Data Capture (REDCap) database, including demographics, mid-upper arm circumference (MUAC), the community where the patient lives, caregiver education and employment, the form of payment for healthcare services, food insecurity, mechanism of injury, time of injury, transfer status to KCMC, and patient outcomes, including in-hospital mortality and morbidity.

## Social Drivers of Health (SDH) variables

SDH factors analyzed were selected a priori based on the Healthcare Information and Management Systems Society (HIMSS) framework [15], encompassing:

- Residence type: categorized as *Moshi Urban* or *outside Moshi Urban (rural/peri-urban)*.

- Health insurance status: insured vs. uninsured.

- Transfer status: direct presentation to KCMC vs. inter-facility transfer (ambulance vs. non-ambulance).

- Caregiver education: no formal education, primary, secondary, or above.

- Household composition: both parents, single parent, or other caregiver.

- Food insecurity: caregiver-reported ability to provide ≥3 daily meals for the child (food-insecure vs. food-secure).

- Payment method: self-pay, government support, or insurance.

These indicators align with SDH domains of economic stability, health system access, and social environment. Additional information regarding the pediatric injury registry data collected is included in the supporting information (S1 File), with SDH factors highlighted.

### Data collection and quality control

The pediatric injury registry data collected have been previously described [16]. Data were collected prospectively by trained research assistants using REDCap tablets at the KCMC EMD [17]. Supervisory and quality checks were performed weekly by the principal investigator (EMK) to ensure accuracy and completeness. Data inconsistencies were verified through review of medical records. All variables were coded and exported to R Statistical Software version 4.3.2 for analysis.

### Statistical analysis

Categorical variables were summarized as frequencies and percentages, and continuous variables were summarized as medians with interquartile ranges (IQRs). Differences in baseline characteristics between outcome groups were initially screened using Chi-square and Wilcoxon rank-sum tests; however, inferences were based exclusively on multivariable modeling.

Two separate models were used:

- Logistic regression to estimate odds ratios (OR) and 95% confidence intervals (CI) for in-hospital mortality.

- Modified Poisson regression with robust standard errors to estimate prevalence ratios (PR) for poor morbidity (GOS-E Peds ≥3).

Model selection rationale: For the mortality outcome, ordinary logistic regression was used because the outcome rate was relatively low (7%). For the poor morbidity outcome (prevalence 38.8%), modified Poisson regression with robust standard errors was selected because it provides more appropriate risk ratio estimates for common outcomes (prevalence >10%) and avoids overestimation of effect sizes that can occur when odds ratios are misinterpreted as risk ratios. Model fit was compared using the Akaike Information Criterion (AIC), and the model with the lowest AIC was selected for final reporting.

Variables with $p < 0.10$ in bivariable screening or theoretical relevance (e.g., age, insurance, residence, food insecurity, transfer status, mechanism of injury) were included in multivariable models. Multicollinearity was assessed using the variance inflation factors (VIF), with all values <5 indicating no problematic collinearity among predictors. Model diagnostics confirmed adequate fit for both mortality and morbidity models. Statistical significance was defined as $p < 0.05$.

### Missing data handling

A complete case analysis was performed, excluding patients missing key SDH or outcome variables (<5% of registry cases). Missingness was determined to be random following cross-variable pattern inspection.

PLOS Global Public Health

## Exploratory analysis

An exploratory post-hoc visualization was performed to illustrate the intersection of food insecurity, insurance status, and morbidity outcomes using a Sankey diagram (Fig 1). This descriptive analysis aimed to demonstrate pathways of social vulnerability and was not included in inferential models.

## Software and reproducibility

All statistical analyses were conducted using R version 4.3.2 (R Foundation for Statistical Computing, Vienna, Austria). Analytical code and non-identifiable data summaries are available upon request from the corresponding or senior author.

## Inclusivity in global research

The supporting information (S1 Checklist) includes additional information regarding the ethical, cultural, and scientific considerations specific to inclusivity in global research.

## Results

### Participant characteristics

From November 2020 through January 2024, a total of 877 pediatric injury patients were enrolled in the registry and met the inclusion criteria for this analysis. The median age was 7 years (IQR 4–12), and 63.3% were male. The majority were uninsured (79.1%), resided outside Moshi Urban (71.3%), and were transferred by ambulance (54%) to KCMC. Over half of patients (61.8%) lived with both parents, and nearly half of caregivers (45.1%) had no formal education. Most families reported being food-secure (61.2%), and most children had normal nutritional status, with a median MUAC of 18.2 cm (IQR 17.0–21.0).

Participant characteristics stratified by mortality status are presented in Table 1, and characteristics stratified by morbidity outcome are presented in Table 2. Among the 877 children, 61 (7%) died during hospitalization (Table 1). Poor

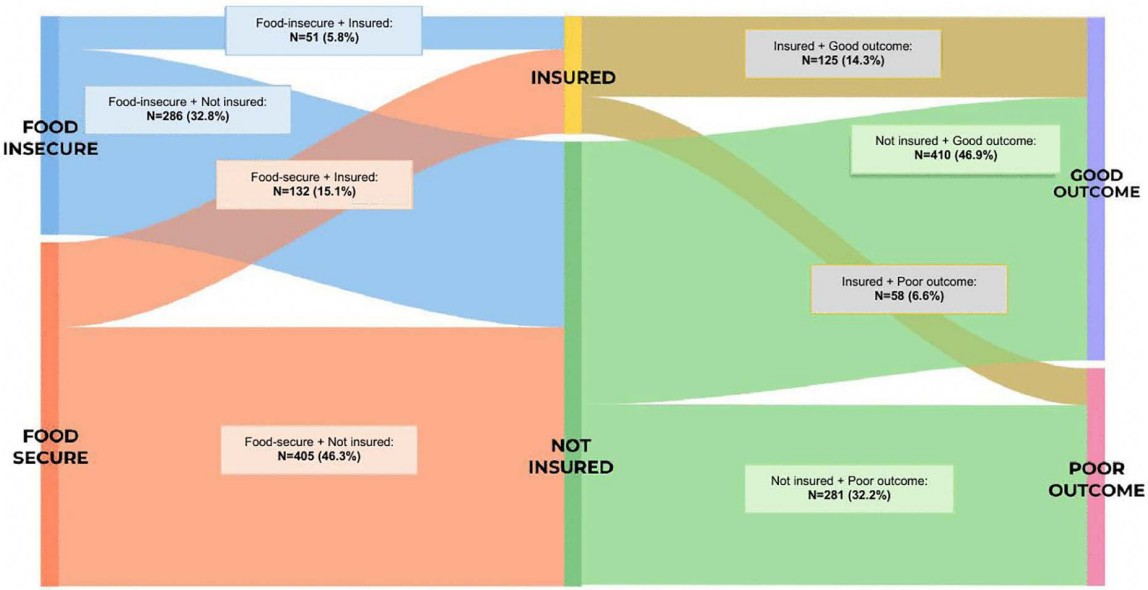

**Fig 1. Sankey diagram showing flows between food security (secure vs. insecure), insurance status (insured vs. uninsured), and outcomes (good vs. poor).** Flow thickness represents category proportions, highlighting differences in outcome distributions across groups.

**Table 1. Socio-demographic characteristics of participants stratified by in-hospital mortality status.**

| Characteristic | Overall N = 877[1] | Deaths N = 61[1] | Survivors N = 816[1] | p-value[2] |
|---|---|---|---|---|
| **Age (Yrs) – Median (IQR)** | 7.0 (4.0, 12.0) | 7.0 (3.0, 12.0) | 7.0 (4.0, 12.0) | 0.818 |
| Unknown | 34 | 4 | 30 | |
| **Sex** | | | | 0.470 |
| Male | 554 (63.3%) | 36 (59.0%) | 518 (63.6%) | |
| Female | 321 (36.7%) | 25 (41.0%) | 296 (36.4%) | |
| Unknown | 2 | 0 | 2 | |
| **Insurance** | | | | 0.012* |
| Insured | 183 (20.9%) | 5 (8.2%) | 178 (21.8%) | |
| Not insured | 694 (79.1%) | 56 (91.8%) | 638 (78.2%) | |
| **Residence** | | | | 0.015* |
| Moshi Urban | 251 (28.7%) | 8 (13.3%) | 243 (29.8%) | |
| Moshi Rural | 227 (25.9%) | 22 (36.7%) | 205 (25.1%) | |
| Other | 398 (45.4%) | 30 (50.0%) | 368 (45.1%) | |
| Unknown | 1 | 1 | 0 | |
| **Live with** | | | | 0.525 |
| Both Parents | 541 (61.8%) | 33 (55.0%) | 508 (62.3%) | |
| Others | 145 (16.6%) | 12 (20.0%) | 133 (16.3%) | |
| Single Parent | 189 (21.6%) | 15 (25.0%) | 174 (21.3%) | |
| Unknown | 2 | 1 | 1 | |
| **Education** | | | | 0.570 |
| None | 392 (45.1%) | 31 (51.7%) | 361 (44.6%) | |
| Primary education | 376 (43.3%) | 23 (38.3%) | 353 (43.6%) | |
| Secondary & Above Education | 101 (11.6%) | 6 (10.0%) | 95 (11.7%) | |
| Unknown | 8 | 1 | 7 | |
| **Transfer Status** | | | | <0.001* |
| Transfer with ambulance (REF) | 473 (54.0%) | 48 (80.0%) | 425 (52.1%) | |
| Transfer without ambulance | 241 (27.5%) | 8 (13.3%) | 233 (28.6%) | |
| Seen first at KCMC | 162 (18.5%) | 4 (6.7%) | 158 (19.4%) | |
| Unknown | 1 | 0 | 1 | |
| **MUAC (cm) – Median (IQR)** | 18.2 (17.0, 21.0) | 17.2 (16.4, 20.2) | 18.5 (17.0, 21.0) | 0.247 |
| Unknown | 63 | 29 | 34 | |
| **Social Worker** | | | | 0.307 |
| No | 782 (89.2%) | 52 (85.2%) | 730 (89.5%) | |
| Yes | 95 (10.8%) | 9 (14.8%) | 86 (10.5%) | |
| **Food Insecurity** | | | | 0.860 |
| Food-insecure | 340 (38.8%) | 23 (37.7%) | 317 (38.8%) | |
| Food-secure | 537 (61.2%) | 38 (62.3%) | 499 (61.2%) | |
| **Tribe of caregiver** | | | | 0.990 |
| Chaga | 349 (39.9%) | 24 (40.7%) | 325 (39.8%) | |
| Others | 393 (44.9%) | 26 (44.1%) | 367 (45.0%) | |
| Pare | 133 (15.2%) | 9 (15.3%) | 124 (15.2%) | |
| Unknown | 2 | 2 | 0 | |
| **Mechanism of Injury** | | | | <0.001* |
| Burn | 100 (11.4%) | 25 (41.0%) | 75 (9.2%) | |
| Fall | 298 (34.0%) | 9 (14.8%) | 289 (35.5%) | |

*(Continued)*

**Table 1.** (Continued)

| Characteristic | Overall N = 877[1] | Deaths N = 61[1] | Survivors N = 816[1] | p-value[2] |
|---|---|---|---|---|
| Others | 184 (21.0%) | 12 (19.7%) | 172 (21.1%) | |
| Road Traffic Injury | 294 (33.6%) | 15 (24.6%) | 279 (34.2%) | |
| Unknown | 1 | 0 | 1 | |

[1]Median (IQR); n (%).

[2]Wilcoxon rank sum test; Pearson's Chi-squared test.

* Indicates statistical significance.

morbidity outcomes (GOS-E Peds ≥3) were observed in 339 (38.8%) cases (Table 2). Burn injuries accounted for the highest proportion of deaths (41%, $n = 25$), followed by road traffic injuries and falls (Table 1).

Regression model assumptions were assessed and found to be satisfied. For logistic regression models, linearity of continuous predictors was verified using restricted cubic splines, and multicollinearity was assessed using variance inflation factors (all VIF 2.5). For modified Poisson models, robust variance estimation was used to account for potential model misspecification.

## Mortality analysis

Findings from the chi-square test indicate a statistically significant association between patient death and health insurance ($p = 0.012$), residence ($p = 0.015$), transfer status ($p < 0.001$), and mechanism of injury ($p < 0.001$) (Table 1).

Table 3 presents unadjusted and adjusted associations between SDH variables and mortality. In crude analysis, mortality was associated with lack of insurance ($p = 0.016$), residence outside Moshi Urban ($p = 0.005, 0.026$), ambulance transfer ($p = 0.002, 0.005$), and burn injury mechanism ($p < 0.001$ for all). After multivariable adjustment, the independent predictors of death that remained significant were:

- Age was associated with mortality (adjusted OR per 1-year increase = 1.27, 95% CI 1.05–1.55, $p = 0.015$).

- Transfer without ambulance was associated with a protective effect on mortality (adjusted OR = 0.18, 95% CI 0.04–0.59, $p = 0.011$).

- Higher MUAC was inversely associated with mortality (adjusted OR = 0.78 per cm, 95% CI 0.65–0.93, $p = 0.008$), suggesting that improved nutritional status conferred protection.

- Food insecurity showed a paradoxical association, with food-secure children demonstrating higher odds of mortality (adjusted OR = 2.62, 95% CI 1.07–7.42, $p = 0.048$). This counterintuitive finding likely reflects unmeasured confounding by injury severity, mechanism, or healthcare-seeking patterns, as discussed in the limitations [18,19].

- Fall mechanism (adjusted OR = 0.08, 95% CI 0.02–0.32, $p < 0.001$) and road traffic injuries (adjusted OR = 0.07, 95% CI 0.02–0.27, $p < 0.001$) were associated with lower odds of mortality compared with burns, reflecting the particularly high case-fatality rate of burn injuries in this setting.

## Morbidity analysis

Chi-square test findings indicate a statistically significant association between morbidity and health insurance ($p = 0.027$), residence ($p = 0.004$), household composition ($p = 0.033$), parental education ($p < 0.001$), transfer status ($p < 0.001$), food insecurity ($p < 0.001$), and mechanism of injury ($p = 0.04$). Wilcoxon rank sum test result indicates a statistically significant

**Table 2. Socio-demographic characteristics of participants stratified by morbidity outcome.**

| Characteristic | Overall N = 874[1] | Good Outcome (GOS-E Peds ≤2) N = 535[1] | Poor Outcome (GOS-E Peds ≥3) N = 339[1] | p-value[2] |
|---|---|---|---|---|
| **Age (Yrs) – Median (IQR)** | 7.0 (4.0, 12.0) | 6.0 (3.0, 10.0) | 9.0 (5.0, 14.0) | <0.001* |
| Unknown | 34 | 25 | 9 | |
| **Sex** | | | | 0.664 |
| Male | 553 (63.4%) | 335 (62.9%) | 218 (64.3%) | |
| Female | 319 (36.6%) | 198 (37.1%) | 121 (35.7%) | |
| Unknown | 2 | 2 | 0 | |
| **Insurance** | | | | 0.027* |
| Insured | 183 (20.9%) | 125 (23.4%) | 58 (17.1%) | |
| Not insured | 691 (79.1%) | 410 (76.6%) | 281 (82.9%) | |
| **Residence** | | | | 0.004* |
| Moshi Urban | 251 (28.8%) | 171 (32.0%) | 80 (23.7%) | |
| Moshi Rural | 224 (25.7%) | 119 (22.2%) | 105 (31.1%) | |
| Other | 398 (45.6%) | 245 (45.8%) | 153 (45.3%) | |
| Unknown | 1 | 0 | 1 | |
| **Live with** | | | | 0.033* |
| Both Parents | 540 (61.9%) | 349 (65.2%) | 191 (56.7%) | |
| Others | 145 (16.6%) | 84 (15.7%) | 61 (18.1%) | |
| Single Parent | 187 (21.4%) | 102 (19.1%) | 85 (25.2%) | |
| Unknown | 2 | 0 | 2 | |
| **Education** | | | | <0.001* |
| None | 391 (45.2%) | 273 (51.5%) | 118 (35.1%) | |
| Primary education | 376 (43.4%) | 216 (40.8%) | 160 (47.6%) | |
| Secondary & Above Education | 99 (11.4%) | 41 (7.7%) | 58 (17.3%) | |
| Unknown | 8 | 5 | 3 | |
| **Transfer status** | | | | <0.001* |
| Transfer with Ambulance | 472 (54.1%) | 253 (47.3%) | 219 (64.8%) | |
| Transfer without Ambulance | 241 (27.6%) | 168 (31.4%) | 73 (21.6%) | |
| Seen first at KCMC | 160 (18.3%) | 114 (21.3%) | 46 (13.6%) | |
| **MUAC (cm) – Median (IQR)** | 18.0 (17.0, 21.0) | 18.0 (17.0, 20.0) | 19.0 (17.0, 23.0) | <0.001* |
| Unknown | 63 | 16 | 47 | |
| **Social Worker** | | | | 0.170 |
| No | 779 (89.1%) | 483 (90.3%) | 296 (87.3%) | |
| Yes | 95 (10.9%) | 52 (9.7%) | 43 (12.7%) | |
| **Food Insecurity** | | | | <0.001* |
| Food-insecure | 337 (38.6%) | 184 (34.4%) | 153 (45.1%) | |
| Food-secure | 537 (61.4%) | 351 (65.6%) | 186 (54.9%) | |
| **Tribe** | | | | 0.764 |
| Chaga | 348 (39.9%) | 209 (39.1%) | 139 (41.2%) | |
| Others | 391 (44.8%) | 245 (45.8%) | 146 (43.3%) | |
| Pare | 133.0 (15.3%) | 81 (15.1%) | 52 (15.4%) | |
| Unknown | 2 | 0 | 2 | |

*(Continued)*

**Table 2.** (Continued)

| Characteristic | Overall N = 874[1] | Good Outcome (GOS-E Peds ≤2) N = 535[1] | Poor Outcome (GOS-E Peds ≥3) N = 339[1] | p-value[2] |
|---|---|---|---|---|
| **Mechanism of Injury** | | | | 0.004* |
| Burn | 100 (11.5%) | 56 (10.5%) | 44 (13.0%) | |
| Fall | 298 (34.1%) | 188 (35.2%) | 110 (32.4%) | |
| Others | 184 (21.1%) | 130 (24.3%) | 54 (15.9%) | |
| Road Traffic Injury | 291 (33.3%) | 160 (30.0%) | 131 (38.6%) | |
| Unknown | 1 | 1 | 0 | |

[1]Median (IQR); n (%).

[2]Wilcoxon rank sum test; Pearson's Chi-squared test.

* Indicates statistical significance.

difference in age (p < 0.001) and in the distribution of MUAC measurements (cm) (p < 0.001) between patients with good versus poor morbidity outcomes (GOS-E Peds score) (Table 2).

In the univariate Poisson regression model for poor morbidity (GOS-E Peds ≥3), several factors remained independently associated with poor recovery (Table 4). However, in multivariate Poisson regression, only age and transfer status were found to be statistically significant. Older age was associated with higher morbidity (APR = 1.09 per year, 95% CI 1.03–1.15, *p* = 0.003, SE 0.028). In contrast, non-ambulance transfer to KCMC was protective (adjusted PR = 0.73, 95% CI 0.54–0.99, *p* = 0.048, SE 0.156), suggesting that direct presentation or informal transport correlated with less severe injuries compared to inter-facility ambulance transfers.

Food insecurity (APR = 1.18, 95% CI 0.95–1.47, p = 0.134) and lack of insurance (APR = 1.21, 95% CI 0.89–1.64, p = 0.228) both demonstrated clinically important trends toward higher prevalence of poor outcomes, but these associations did not reach statistical significance after controlling for confounders including age, MUAC, transfer status, and injury mechanism. No consistent differences were observed in caregiver education or household composition after controlling for other SDH variables.

## Exploratory visualization: Pathways of social vulnerability

Fig 1 illustrates the interconnections among food insecurity, insurance status, and morbidity outcomes using a Sankey diagram, demonstrating the gradient of social vulnerability across recovery trajectories. Among food-secure and uninsured patients (n = 405), 383 (94.6%) experienced good recovery (GOS-E Peds ≤2), while 22 (5.4%) experienced poor outcomes (GOS-E Peds ≥3). In contrast, among food-insecure and uninsured patients (n = 286), 27 (9.4%) experienced good recovery, while 259 (90.6%) experienced poor outcomes. Although descriptive, this visualization highlights how the convergence of food insecurity and lack of insurance channels children toward poorer outcomes, underscoring the cumulative effect of social risk on post-injury recovery.

## Summary of findings

In summary, older age, inter-facility ambulance transfer (as a marker of injury severity), lower MUAC, and burn injury emerged as the strongest predictors of in-hospital mortality. Regarding morbidity, older age and inter-facility ambulance transfer were consistent independent risk factors. Food insecurity remained a cross-cutting vulnerability affecting both survival and functional recovery, though the direction and statistical significance of associations differed between outcomes and require cautious interpretation given potential unmeasured confounding. These findings suggest that both socioeconomic and system-level factors influence pediatric trauma trajectories in Northern Tanzania.

 

**Table 3. Univariate and multivariate logistic regression models for in-hospital mortality.**

| Characteristics | OR (95%CI) | P-value | AOR (95%CI) | P-value |
|---|---|---|---|---|
| **Age** (Yrs) | 1.00 (0.94-1.06) | 0.928 | 1.27 (1.05-1.55) | 0.015* |
| **Sex** | | 0.471 | | 0.711 |
| Male | REF | | REF | |
| Female | 1.22 (0.71-2.05) | | 1.17 (0.49, 2.69) | |
| **Insurance** | | 0.016* | | 0.258 |
| Insured | REF | | REF | |
| Not insured | 3.13 (1.36-9.06) | | 2.59 (0.60-18.5) | |
| **Residence** | | | | |
| Moshi Urban | REF | | REF | |
| Moshi Rural | 3.26 (1.48 7.95) | 0.005* | 1.54 (0.46, 5.72) | 0.495 |
| Other | 2.48 (1.17 5.88) | 0.026* | 0.91 (0.28, 3.38) | 0.877 |
| **Live with** | | | | |
| Both Parents | REF | | REF | |
| Others | 1.40 (0.67-2.69) | 0.349 | 1.17 (0.39-3.12) | 0.758 |
| Single Parent | 1.33 (0.69-2.46) | 0.382 | 0.64 (0.18-1.82) | 0.433 |
| **Education** | | | | |
| None | REF | | REF | |
| Primary education | 0.76 (0.43-1.32) | 0.333 | 0.61 (0.18-2.09) | 0.422 |
| Secondary & Above Education | 0.74 (0.27-1.70) | 0.505 | 1.13 (0.15-9.36) | 0.907 |
| **Transfer status** | | | | |
| Transfer with Ambulance | REF | | REF | |
| Transfer without Ambulance | 0.30 (0.13-0.62) | 0.002* | 0.18 (0.04-0.59) | 0.011* |
| Seen first at KCMC | 0.22 (0.07-0.56) | 0.005* | 0.21 (0.03-0.96) | 0.075 |
| **MUAC** | 0.78 (0.85-1.04) | 0.286 | 0.78 (0.65-0.93) | 0.008* |
| **Social Worker** | | | | |
| No | REF | | REF | |
| Yes | 1.47 (0.66-2.95) | 0.310 | 0.73 (0.18-2.28) | 0.622 |
| **Food Insecurity** | | | | |
| Food-insecure | REF | | REF | |
| Food-secure | 1.05 (0.62-1.82) | 0.86 | 2.62 (1.07-7.42) | 0.048* |
| **Tribe of the caregiver** | | | | |
| Chaga | REF | | REF | |
| Others | 0.96 (0.54-1.71) | 0.887 | 1.07 (0.41-2.79) | 0.889 |
| Pare | 0.98 (0.42-2.10) | 0.966 | 0.46 (0.09-1.72) | 0.283 |
| **Mechanism of injury** | | | | |
| Burn | REF | | REF | |
| Fall | 0.09 (0.04- 0.20) | <0.001* | 0.08 (0.02-0.32) | <0.001* |
| Others | 0.21 (0.10-0.43) | <0.001* | 0.19 (0.05- 0.75) | 0.020* |
| Road Traffic Injury | 0.16 (0.08- 0.32) | <0.001* | 0.07 (0.02- 0.27) | <0.001* |

* Indicates statistical significance.

**Table 4. Univariate and multivariate modified Poisson regression models for the prevalence ratio of morbidity/poor outcome (GOS-E Peds ≥3).**

| Characteristics | PR (95%CI) | P-value | Standard error | APR (95%CI) | P-value | Standard error |
|---|---|---|---|---|---|---|
| **Age (Yrs)** | 1.06 (1.03, 1.08) | <0.001* | 0.011 | 1.09 (1.03, 1.15) | 0.003* | 0.028 |
| **Sex** | | | | | | 0.131 |
| Male | REF | | | REF | | |
| Female | 0.96 (0.77, 1.20) | 0.7 | 0.113 | 1.04 (0.80, 1.34) | 0.755 | |
| **Insurance** | | | | | | 0.185 |
| Insured | REF | | | REF | | |
| Not insured | 1.28 (0.98, 1.72) | 0.084 | 0.084 | 1.05 (0.74, 1.53) | 0.789 | |
| **Residence** | | | | | | |
| Moshi Urban | REF | | | REF | | |
| Moshi Rural | 1.47 (1.10, 1.97) | 0.009* | 0.148 | 1.12 (0.79, 1.58) | 0.535 | 0.176 |
| Other | 1.21(0.92, 1.59) | 0.2 | 0.138 | 0.87 (0.63, 1.22) | 0.406 | 0.169 |
| **Live with** | | | | | | |
| Both Parents | REF | | | REF | | |
| Others | 1.19 (0.88, 1.58) | 0.2 | 0.147 | 1.05 (0.76, 1.43) | 0.768 | 0.163 |
| Single Parent | 1.29 (0.99, 1.65) | 0.054* | 0.130 | 1.09 (0.81, 1.45) | 0.55 | 0.148 |
| **Education** | | | | | | |
| None | REF | | | REF | | |
| Primary education | 1.41 (1.11, 1.79) | 0.005* | 0.121 | 1.01 (0.69, 1.47) | 0.964 | 0.192 |
| Secondary & Above Education | 1.94 (1.41, 2.65) | <0.001* | 0.160 | 1.14 (0.63, 2.10) | 0.670 | 0.308 |
| **Transfer Status** | | | | | | |
| Transfer with Ambulance | REF | | | REF | | |
| Transfer without Ambulance | 0.65 (0.50-0.85) | 0.002* | 0.135 | 0.73 (0.54-0.99) | 0.048* | 0.156 |
| Seen first at KCMC | 0.62 (0.45- 0.84) | 0.003* | 0.162 | 0.70 (0.47-1.02) | 0.068 | 0.199 |
| **MUAC** | 1.05 (1.02, 1.07) | <0.001* | 0.012 | 0.97 (0.92, 1.01) | 0.163 | 0.025 |
| **Social Worker** | | | | | | |
| No | REF | | | REF | | |
| Yes | 1.19 (0.85, 1.62) | 0.3 | 0.163 | 1.14 (0.78, 1.62) | 0.470 | 0.186 |
| **Food Insecurity** | | | | | | |
| Food-insecure | REF | | | REF | | |
| Food-secure | 0.76 (0.62, 0.95) | 0.013* | 0.109 | 0.86 (0.67, 1.10) | 0.222 | 0.126 |
| **Tribe** | | | | | | |
| Chaga | REF | | | REF | | |
| Others | 0.93 (0.74, 1.18) | 0.6 | 0.118 | 1.09 (0.82, 1.45) | 0.543 | 0.145 |
| Pare | 0.98 (0.71, 1.34) | 0.9 | 0.162 | 1.07 (0.73, 1.55) | 0.713 | 0.190 |
| **Mechanism of injury** | | | | | | |
| Burn | REF | | | REF | | |
| Fall | 0.84 (0.60-1.20) | 0.325 | 0.178 | 0.86 (0.52, 1.51) | 0.592 | 0.273 |
| Others | 0.67 (0.45-1.00) | 0.046* | 0.203 | 0.68 (0.39, 1.23) | 0.191 | 0.293 |
| Road Traffic Injury | 1.02 (0.73-1.45) | 0.896 | 0.174 | 0.88 (0.53, 1.53) | 0.631 | 0.271 |

* Indicates statistical significance.

## Discussion

This study examined the association between selected SDH and pediatric injury outcomes at a tertiary hospital in Northern Tanzania. Our findings highlight that older age, inter-facility ambulance transfer, lower MUAC, and burn injuries were independently associated with higher mortality. Regarding morbidity, older age and ambulance transfer were associated with increased poor functional outcomes. Food insecurity and lack of insurance demonstrated clinically important associations with both survival and recovery outcomes, though statistical significance and direction varied by outcome. These findings align with reports on the burden of SDH factors associated with increased risk of injury in LMICs [1,20–22]. This analysis provides insight into factors that could be addressed through in-hospital and community-level interventions.

### Unmodifiable factors

The relationship between SDH and the health outcomes of children experiencing injuries is influenced by factors such as sex and ethnicity. Sex is an important determinant of injury, with males at higher risk than females, as our findings demonstrate [23]. This may be due to differences in behavior, access to resources, and societal expectations. Ethnicity can also impact the likelihood of injury, as certain ethnic groups may experience higher levels of social exclusion and lack access to resources and services. For example, rural and indigenous populations in many LMICs experience higher rates of injury compared to non-indigenous populations [1].

### Health equity and insurance access

Lack of health insurance emerged as one of the strongest predictors of mortality. In Tanzania, where insurance coverage is predominantly purchased out-of-pocket, insurance status likely reflects broader socioeconomic disadvantage. Similar findings have been observed in LMIC settings, where uninsured children are less likely to receive timely interventions and experience higher rates of preventable death [4,24–27]. The relationship between insurance and outcomes underscores inequities in healthcare access and affordability, aligning with SDG 3.8 ("Achieve universal health coverage") [12]. Policies that expand insurance coverage or public subsidies for pediatric trauma care may mitigate these disparities.

### Transportation and trauma system challenges

Ambulance transport was independently associated with both increased mortality and morbidity. In the Tanzanian context, ambulances are typically used for interfacility transfers rather than prehospital scene responses. Thus, ambulance utilization likely identifies patients with more severe injuries rather than implying inadequate care. However, deficiencies in prehospital stabilization and delayed transport could contribute to worse outcomes. Previous trauma registry studies in Malawi and Ethiopia have similarly linked ambulance transport to higher injury severity and delayed definitive care [28,29]. Ambulances are usually prioritized for transporting sicker patients, whereas less severe injuries might be transported by public transportation or private cars. Thus, our findings of increased morbidity and mortality in children transported by ambulance are expected. Yet another possible contributor to poorer outcomes in patients transported by ambulance could be a lack of formalized transport care, leading to sicker patients upon arrival. Strengthening prehospital triage and establishing formal EMS are essential to improving survival among critically injured children.

### Food security and child vulnerability

Food insecurity revealed complex associations with outcomes that require careful interpretation. The paradoxical finding that food-secure children showed higher mortality odds in adjusted models is counterintuitive and likely reflects unmeasured confounding by injury severity, mechanism, or socioeconomic gradient reversals within this cohort [18,19].

Recent evidence confirms that food insecurity is a major driver of adverse pediatric health outcomes, including increased risk of mortality, poor recovery, and long-term disability [18,19]. In low-income countries, undernutrition accounts

for nearly half of all deaths among children under 5 years of age, with increased mortality from infectious diseases and impaired physiologic resilience following injury [19]. The relationship between food insecurity and injury outcomes is multi-factorial, with confounding by injury severity, socioeconomic status, and healthcare access potentially producing counter-intuitive findings, as observed in our study [18,19].

Several mechanisms may explain our paradoxical mortality finding. Children from food-secure households in our cohort may have presented with more severe injuries due to differences in risk exposure, healthcare-seeking behavior, or referral patterns from district hospitals. Additionally, measurement limitations—including the use of a single-item, non-validated food insecurity measure and the bidirectional nature of food insecurity—further complicate interpretation [19]. Collider bias may also contribute: conditioning on hospital admission in a tertiary referral center may induce spurious associations if both food security and injury severity independently influence the likelihood of reaching KCMC.

Despite the paradoxical mortality finding, food insecurity remained associated with functional impairment in the expected direction (though not statistically significant), reflecting chronic nutritional deficits that limit physiologic resilience and recovery [30,31]. Higher MUAC, an objective marker of nutritional status, was independently protective against mortality, supporting the biological plausibility that better nutrition improves injury outcomes [19].

Integrating robust, multidimensional assessments of food security and controlling for key confounders, including injury severity scores, is essential for future research [18,19]. Nonetheless, addressing food insecurity through targeted nutrition and social protection interventions remains critical to improving pediatric injury outcomes in LMICs(18, 19). Integrating nutritional assessment and food support into pediatric trauma care could reduce post-injury disability. Addressing food insecurity aligns directly with SDG 2 ("Zero Hunger") and SDG 3 ("Good Health and Well-being"), emphasizing the interdependence of nutrition and health outcomes in vulnerable populations [12,19].

### Community context and access to care

Children living in rural or peri-urban communities outside Moshi Urban exhibited poorer outcomes, consistent with evidence that geographic inequities restrict timely access to healthcare outcomes [2,20–22,24,32]. Distance to tertiary facilities, poor road infrastructure, and limited referral coordination prolong injury-to-care intervals [32]. Targeted investment in district-level trauma capacity and referral networks could alleviate the disproportionate burden of morbidity and mortality among rural children. This aligns with SDG 10 ("Reduced Inequalities") through the decentralization of emergency services [12].

### Linkages to the Sustainable Development Goals (SDGs)

The findings of this study intersect with several SDGs, particularly SDG 3 (Good Health and Well-being), SDG 2 (Zero Hunger), and SDG 10 (Reduced Inequalities) [12]. By illustrating the impact of socioeconomic inequities on pediatric injury outcomes, our data highlight the need for multi-sectoral approaches that integrate social protection, nutrition, and health system strengthening. SDH-sensitive injury registries could provide an evidence base for monitoring progress toward these global targets [12].

### Implications and future directions

Our findings underscore the importance of integrating SDH surveillance within pediatric trauma systems. Systematic collection of SDH indicators could help identify children at risk for poor outcomes and guide resource allocation. Future research should examine how targeted interventions — such as subsidized insurance, nutritional support, and decentralized trauma care — affect morbidity trajectories. Additionally, developing validated tools for SDH assessment tailored to the Tanzanian context could strengthen local data systems and support equitable implementation of health policies.

## Limitations

This study has several important limitations. Selection and survival bias inherent to a single-center, tertiary referral hospital design likely resulted in underestimation of mortality and its associated risk factors, as the most severely injured children—particularly from rural areas—may have died before reaching KCMC or during transport, a well-described limitation of trauma registries in LMICs. The single-center design also limits generalizability to other trauma care settings. A critical methodological limitation is the absence of validated injury severity measures, which precludes adjustment for baseline anatomic and physiologic injury burden and introduces potential confounding into observed associations between SDH and outcomes. SDH data were self-reported and subject to recall and social desirability bias, and food insecurity was measured using a single-item proxy rather than a validated multidimensional scale, potentially leading to misclassification and attenuation or distortion of true associations. Finally, the cross-sectional design precludes causal inference, and residual confounding from unmeasured variables such as prehospital time, referral care quality, and comorbidities may persist despite multivariable adjustment. Collectively, these limitations suggest that observed associations should be interpreted as associative rather than causal; nevertheless, our findings align with a growing body of literature demonstrating that socioeconomic inequities shape pediatric injury outcomes in LMICs and underscore the importance of integrating SDH surveillance into trauma care systems.

## Conclusions

In this secondary analysis of a prospective pediatric injury registry, older age, inter-facility ambulance transfer, lower MUAC, food security, and burn injuries were independently associated with increased mortality. In comparison, older age and ambulance transfer were associated with greater morbidity. Food insecurity and lack of insurance remained cross-cutting vulnerabilities. Addressing these SDH through integrated health and social policy approaches could improve pediatric injury outcomes and advance equity in LMIC settings. Establishing SDH-informed injury registries may be a crucial step toward achieving SDGs focused on child health, nutrition, and reducing inequality.

## Supporting information

**S1 File. Registry Intake.** Additional information regarding the pediatric injury registry data collected with SDH factors highlighted.
(PDF)

**S1 Checklist. Inclusivity in global research.** Additional information regarding the ethical, cultural, and scientific considerations specific to inclusivity in global research.
(DOCX)

## Acknowledgments

The authors would like to thank the pediatric injury patients and their caregivers who participated in the KCMC pediatric injury registry, as well as the research assistants and clinical teams who supported data collection and patient care.

## Author contributions

**Conceptualization:** Natalie J Tedford, Modesta Mitao, Timothy Antipas Peter, Linda Minja, Blandina T. Mmbaga, Elizabeth M. Keating.

**Data curation:** Getrude Nkini, Elizabeth M. Keating.

**Formal analysis:** Modesta Mitao, Timothy Antipas Peter, Gabriele Nascimento de Oliveira, Joao R.N. Vissoci.

**Funding acquisition:** Catherine Staton, Blandina T. Mmbaga, Elizabeth M. Keating.

**Investigation:** Natalie J Tedford, Elizabeth M. Keating.

**Methodology:** Natalie J Tedford, Modesta Mitao, Timothy Antipas Peter, Joao R.N. Vissoci, Elizabeth M. Keating.

**Project administration:** Getrude Nkini, Elizabeth M. Keating.

**Resources:** Catherine Staton, Blandina T. Mmbaga, Elizabeth M. Keating.

**Software:** Modesta Mitao, Timothy Antipas Peter, Linda Minja, Gabriele Nascimento de Oliveira, Elizabeth M. Keating.

**Supervision:** Catherine Staton, Joao R.N. Vissoci, Blandina T. Mmbaga, Elizabeth M. Keating.

**Validation:** Timothy Antipas Peter, Joao R.N. Vissoci.

**Visualization:** Timothy Antipas Peter, Gabriele Nascimento de Oliveira.

**Writing – original draft:** Natalie J Tedford, Elizabeth M. Keating.

**Writing – review & editing:** Natalie J Tedford, Modesta Mitao, Timothy Antipas Peter, Linda Minja, Gabriele Nascimento de Oliveira, Raven Mingo, Getrude Nkini, Catherine Staton, Joao R.N. Vissoci, Blandina T. Mmbaga, Elizabeth M. Keating.

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
