## [Decision Letter · Decision Letter 0]

29 Sep 2025

PGPH-D-25-02665

Exploring the relationship between social determinants of health and pediatric injury outcomes at a Northern Tanzania tertiary referral hospital

Dear Dr. Tedford,

Thank you for submitting your manuscript to PLOS Global Public Health. After careful consideration, we feel that it has merit but does not fully meet PLOS Global Public Health’s publication criteria as it currently stands. Therefore, we invite you to submit a revised version of the manuscript that addresses the points raised during the review process.

We look forward to receiving your revised manuscript.

Kind regards,

Hani Mowafi, M.D., M.P.H.

Academic Editor

Journal Requirements:

1. We noticed you have some minor occurrence of overlapping text with the following previous publication(s), which needs to be addressed:

- https://doi.org/10.1371/journal.pgph.0000657

In your revision ensure you cite all your sources (including your own works), and quote or rephrase any duplicated text outside the methods section. Further consideration is dependent on these concerns being addressed.

i. State the initials, alongside each funding source, of each author to receive each grant.

ii. State what role the funders took in the study. If the funders had no role in your study, please state: “The funders had no role in study design, data collection and analysis, decision to publish, or preparation of the manuscript.”

3. Please ensure that your Ethics Statement is available in its entirety at the beginning of your Methods section, under a subheading 'Ethics Statement'. It must include:

1) The name(s) of the Institutional Review Board(s) or Ethics Committee(s)

2) The approval number(s), or a statement that approval was granted by the named board(s)

3) (for human participants/donors) - A statement that formal consent was obtained (must state whether verbal/written) OR the reason consent was not obtained (e.g. anonymity).

NOTE: If child participants, the statement must declare that formal consent was obtained from the parent/guardian.

4. Please provide separate figure files in .tif or .eps format.

5. Please provide a complete Data Availability Statement in the submission form, ensuring you include all necessary access information or a reason for why you are unable to make your data freely accessible. If your research concerns only data provided within your submission, please write "All data are in the manuscript and/or supporting information files" as your Data Availability Statement.

6. Some material included in your submission may be copyrighted. According to PLOS’s copyright policy, authors who use figures or other material (e.g., graphics, clipart, maps) from another author or copyright holder must demonstrate or obtain permission to publish this material under the Creative Commons Attribution 4.0 International (CC BY 4.0) License used by PLOS journals. Please closely review the details of PLOS’s copyright requirements here: PLOS Licenses and Copyright. If you need to request permissions from a copyright holder, you may use PLOS's Copyright Content Permission form.

Potential Copyright Issues:

a. We do not publish any copyright or trademark symbols that usually accompany proprietary names, eg (R), (C), or TM (e.g. next to drug or reagent names). Therefore please remove all instances of trademark/copyright symbols throughout the text, including ‘REDCap®’ on page 8.

Additional Editor Comments (if provided):

Reviewers' comments:

Reviewer's Responses to Questions

**Comments to the Author**

1. Does this manuscript meet PLOS Global Public Health’s publication criteria?

Reviewer #1: Partly

Reviewer #2: Yes

Reviewer #3: Partly

2. Has the statistical analysis been performed appropriately and rigorously?

Reviewer #1: No

Reviewer #2: Yes

Reviewer #3: No

3. Have the authors made all data underlying the findings in their manuscript fully available (please refer to the Data Availability Statement at the start of the manuscript PDF file)?

Reviewer #1: Yes

Reviewer #2: Yes

Reviewer #3: Yes

4. Is the manuscript presented in an intelligible fashion and written in standard English?

Reviewer #1: Yes

Reviewer #2: No

Reviewer #3: Yes

Reviewer #1: Please review the results section. Make sure data from table 2a and table 4 are accurately reported in terms of p-values and analysis. The paragraph introducing the Sankey Diagram starting from "Our analysis showed an underlying social vulnerability concept that defined the association with “surviving” and “poor outcomes,” represented by food insecurity..." should be re-visited. The concepts should be made clearer to the reader. It could use clarifications.

Reviewer #2: I have attached my detailed review for your consideration. Major comments need to be addressed before the paper becomes ready for publication especially those related to study design and reporting of results. Please check out the attached document which has all of my comments.

Reviewer #3: Review of: PGPH-D-25-02665

Summary of the study:

his study is a secondary data analysis of an established registry from a tertiary referral hospital in Northern Tanzania. The authors aim to identify social determinants of health associated with increased morbidity and mortality among pediatric patients who sustained injuries and were transferred to Kilimanjaro Christian Medical Centre.

Title:

1- “Pediatric injury outcome” is a very broad term. Please consider rewriting the title into a more specific one. Even though the study is exploratory, “social determinants of health” is too broad.

Abstract:

1- It’s better to report the multivariable analysis in the abstract rather than reporting the univariable analysis with respect to mortality.

2- What were the variables adjusted for in the “Adjusted Poisson Regression” model for assessing the association with morbidity?

3- Remove the reference to “Figure 1” from the abstract.

4- Under the “Conclusion,” it would be better to specify the direction of association between the levels of the independent factor and the outcome. In addition, only report the factors from the multivariable regression in your conclusions.

Introduction

1- Line 62 – 67: The link between EXPERIENCING injuries and the mentioned factors is unclear.

2- Line 73 & 74: What are these disadvantages? And how did you relate this to your topic under study?

3- “SDH and pediatric trauma” is a very broad phrase and could be answered by several study designs based on your main objective. Please be more concise.

4- Line 77 & 78: Clarify what you’re referring to as health inequality monitoring and keep it within the scope of your study.

5- Line 77 – 80: Understanding the complex interplay between SDH and the health outcomes of injured children doesn’t help you identify effective strategies to prevent injuries from happening. To prevent injuries in the first place, you need to study the factors associated with their incidence. In this case, you must approach your study design differently, such as conducting a cohort study or a case-control study. On the other hand, to prevent the increased severity and negative outcomes of an injury, you focus on the factors associated with post-injury trajectories. Please revise this section accordingly.

6- Line 82: The gap doesn’t match your aim.

7- Lines 85 & 86: On what basis were the SDH factors selected?

Methods

1- Identify the study as a secondary data analysis study since the registry wasn’t created initially to answer the research question under your current submission.

2- Outcomes – mortality and morbidity – should be defined in detail in a separate section, “Study Outcome Variables.”

3- Lines 116 – 120 should reflect that data collection for the registry and not this study was conducted at KCMC.

4- It would be better to describe the registry, data collectors' strategies, research assistants following up with patients’ care, variables collected, and follow-ups in the main manuscript text and keep the details in the supplementary material.

5- Lines 152 – 155: Move to “Statistical Analysis” section.

6- Under the “Statistical Analysis” section, identify between parentheses the variables’ names and measurement scales for each of the categorical and continuous types.

7- You can assess differences at the bivariable level using the Chi-square and Wilcoxon rank sum tests, but not to report adjusted measures of association. Please revise accordingly.

8- Lines 163 – 165: Which models are you referring to? And on what basis did you decide to compare the models?

9- What tools were used to assess independent variables? Validity of the tools? The reader expects to find this information in the Methods without the need to go back to the original data collection methods and details of the registry. More details could be accessed through the reference provided, but such important information on tools is needed under the Methods section.

10- What is your definition of Survivor versus non-Survivor? Time-frame?

11- The definition and classification of Good or Poor outcomes must be included in the Methods section, as this is currently missing from your submission. At present, the first mention appears in the Table and subsequently in the Results. Please revise accordingly.

12- Report on the variables with missing data and the respective percentages.

13- Define the timeframe you’re referring to for mortality and morbidity assessments.

14- Did you rely on GOS-E only for assessing morbidity (i.e., specifically TBI)?

15- Which modified Poisson regression was utilized?

Results

1- Do not report univariable analysis (i.e., chi-squared test and Wilcoxon rank test) as statistical tests to conclude associations. Please use them for screening potential co-variables and focus on multivariable analysis.

2- In the adjusted analysis, how did you treat the “Mechanism of Injury” groups? How did you enter it in the model, and how did you analyze the different groups?

3- Lines 219 – 221: The age results reporting is unclear. Please revise.

4- The titles of the tables are misleading. Consider revising them by indicating the outcome variable clearly (survivors is misleading), the number of participants, the region(s), and the year(s).

5- Lines 244 – 256: This is exploratory analysis or post-hoc analysis. You should mention this under the methods section and categorize it under a separate section in the Results.

6- Lines 244 – 256: Where are your comparison methods and results?

7- I don’t understand the need for a Sankey diagram. The reasoning is unclear to me. It is a visualization tool. How has it solved the problem you’re referring to?

8- Don’t summarize, interpret, or discuss any of your results/findings in the Results section. Move it to the Discussion (e.g., Our analysis showed; indicating a disparity)

Discussion

1- Lines 271 and 272: The significance here should be very specific to your results and not a general conclusion.

2- Lines 274 - 281: The reference to the literature must be cited next to it and not next to your results. In addition, sex and ethnicity are two different factors. Present them in two separate paragraphs or two separate ideas. Provide more examples of how each factor affects, while referencing relevant literature or sources.

3- Lines 283 – 285: Use parallel reporting when comparing with the literature: present exposure and outcome in the same order across studies to facilitate comparison. For example: “Lack of health insurance was associated with increased morbidity and mortality in our study, consistent with prior findings linking lack of insurance to poor health outcomes.” Avoid reversing the structure (e.g., “Poor outcomes were previously linked to lack of insurance”) as it disrupts the comparative flow, whenever possible.

4- Delve further into the potential mechanisms underlying the observed associations, and avoid broad or generic explanations such as “Several studies have highlighted the significant role SDH has on the health outcomes of children experiencing injuries in LMICs, 294 noting that poverty was associated with sustaining more severe injuries, having more extended hospital stays, and experiencing poorer outcomes.” Instead, emphasize the specific pathways, the how and why SDH influences these outcomes.

5- Lines 298 – 309: Support this finding by presenting the severity status of your study patients and emphasizing this in your Results section.

6- Lines 321 – 327: Such conclusions require path and moderator analysis.

Conclusions

This is too general. Please consider grounding it in your specific results to enhance clarity and the reader’s understanding of the study’s specific significance and relevance.

The manuscript requires revision for English language and sentence flow/structure.

**Do you want your identity to be public for this peer review?** For information about this choice, including consent withdrawal, please see our Privacy Policy

Reviewer #1: No

Reviewer #2: No

Reviewer #3: No

---

## [Decision Letter · Decision Letter 1]

5 Jan 2026

PGPH-D-25-02665R1

Social drivers and pediatric injury outcomes in Northern Tanzania: A prospective pediatric injury registry secondary analysis

Dear Dr. Tedford,

Thank you for submitting your manuscript to PLOS Global Public Health. After careful consideration, we feel that it has merit but does not fully meet PLOS Global Public Health’s publication criteria as it currently stands. Therefore, we invite you to submit a revised version of the manuscript that addresses the points raised during the review process.

Please make suggested revisions before resubmission. In particular, there are important revisions needed in the presentation of results.

We look forward to receiving your revised manuscript.

Kind regards,

Hani Mowafi, M.D., M.P.H.

Academic Editor

Journal Requirements:

Additional Editor Comments (if provided):

Please make suggested revisions before resubmission. In particular, there are important revisions needed in the results. Section that are required for clarification of the data being presented.

Reviewers' comments:

Reviewer's Responses to Questions

**Comments to the Author**

Reviewer #1: All comments have been addressed

Reviewer #3: (No Response)

publication criteria?

Reviewer #1: Partly

Reviewer #3: (No Response)

3. Has the statistical analysis been performed appropriately and rigorously?

Reviewer #1: No

Reviewer #3: (No Response)

4. Have the authors made all data underlying the findings in their manuscript fully available (please refer to the Data Availability Statement at the start of the manuscript PDF file)?

Reviewer #1: Yes

Reviewer #3: (No Response)

5. Is the manuscript presented in an intelligible fashion and written in standard English?

Reviewer #1: Yes

Reviewer #3: (No Response)

Reviewer #1: The Results section is still not reported accurately.

In the section of participants characteristics, please indicate what table the reader should refer to in order to verify results.

In the Mortality analysis section, the P-Value for insurance is reported to be (0.027) which is inaccurate as it reflects the results from table 2 for morbidity outcomes. The actual P-value here is 0.012 (from Table 1). Please make sure to review all the reported values again.

Results reported from Table 3 also require revision in terms of analysis.

Report numbers in this section: Food insecurity and lack of insurance both trended toward higher prevalence of poor

outcomes, but these associations did not reach statistical significance after controlling for

confounders. No consistent differences were observed in caregiver education or household

composition after controlling for other SDH variables.

Please review reported results of the Sanke Diagram. The title of Figure 1 should be placed with the figure not separately.

Reviewer #3: Minor Re-Revision Comments:

1. The definition of "injury" should not be placed under the "Study Participants" section. Please relocate it to the appropriate section.

2. There is inconsistency in your terminology; you use "food security" in some instances and "food insecurity" in others. Please maintain consistent terminology throughout the entire manuscript, focusing on the term that aligns with its role as a predictor.

3. You conducted assumption testing. Its results must be included in the results section, even if only briefly.

4. You reported the results of the modified Poisson regression, but you stated: "AIC was selected from an ordinary logistic regression model compared with a modified Poisson model." Please clarify what you meant by this statement.

5. The titles of the tables are misleading and lack informativeness. Please revise them and use "deaths" or "mortality" instead of "non-survivor."

6. There appears to be an issue with some variables in the reported tables. For example, in Table 3, the adjusted odds ratio (AOR) for mortality regarding the predictor "food insecurity" is listed as 2.62 for the "No" versus "Yes" response. This suggests that "food insecurity" is protective, which is inaccurate. Please revisit this and ensure the categorization and labeling of your variables, including their corresponding reference levels in the analysis, are correct.

7. In the first paragraph of the Discussion section, please replace the phrases "In comparison" and "meaningful relationship."

8. Emphasize more how the limitations of your study might have distorted or affected the measures of association reported in the results.

**Do you want your identity to be public for this peer review?** For information about this choice, including consent withdrawal, please see our Privacy Policy

Reviewer #1: No

Reviewer #3: No

---

## [Editor Report · Decision Letter 2]

26 Jan 2026

Social drivers and pediatric injury outcomes in Northern Tanzania: A prospective pediatric injury registry secondary analysis

PGPH-D-25-02665R2

Dear Dr Tedford,

We are pleased to inform you that your manuscript 'Social drivers and pediatric injury outcomes in Northern Tanzania: A prospective pediatric injury registry secondary analysis' has been provisionally accepted for publication in PLOS Global Public Health.

Best regards,

Hani Mowafi, M.D., M.P.H.

Academic Editor